# Hitchhiking Experiences and Perception of Affective Label Polarity in Social Networking Sites—Potential Memetic Implications for Digital Visual Content Management

Krzysztof Stepaniuk [1,*] and Anna Sturgulewska [2]

1   Department of Tourism and Marketing, Faculty of Engineering Management, Bialystok University of Technology, 16-001 Kleosin, Poland
2   Augustow Tourist Organization, 16-300 Augustow, Poland; it@urzad.augustow.pl
*   Correspondence: k.stepaniuk@pb.edu.pl

**Abstract:** Digital Visual Content (DVC), as sets of meanings, constitutes a pictorial representation of phenomena and may lead users of Social Networking Sites (SNS) to specific behavioral activities, i.e., connected with the expression of emotions. The goal of this research paper was to create a methodology of analysis and visualization of SNS users' emotional responses to the hitchhiker archetype, built on the basis of a semantic decomposition of photos shared by hitchhikers in a closed group of Facebook users. Hitchhiker, as an archetype of authentic and sustainable tourism, was used as an illustrative example. ABSA (Aspect-Based Sentiment Analysis) methodology was used for a semantic decomposition of 300 randomly selected photos, applying the theoretical framework of information ecology. As a result, collections of single meanings (aspects) were distinguished and grouped into four main denotation groups: "Hitchhiker environment", "Activities of hitchhikers", "Season" and "Heritage". Having created a holistic view of the archetype from aspects appearing at different frequencies, it was assumed that the archetype could be viewed as a meme. Given this memetic archetype and the emotional response associated with it, it could be considered a subject of cultural selection, conditioned by the influence of several factors, e.g., user demographics. The results allow for the development of the premises for research related to the dynamics, evolutionary analysis, and visualization of memes as well as emotional responses of visual content recipients.

**Keywords:** social networking sites; hitchhiker; emotional perception; visualization; archetypes; meme theory; digital visual content management

## 1. Introduction

Sharing photos on Social Networking Sites (SNS) is one of the most frequent activities of their users [1]. The process of photograph sharing—as an element of User Generated Content (UGC) and specific forms of a message—constitutes an element of the SNS' communication strategy. Such a process enables building an image desired by the source [2], among others, due to stimulating the creation of positive emotions, impressions [3], and social approval [4]. The properties indicated above tend to refer to individual photographs as information means in terms of information ecology (IE). IE is defined as "a system of people, practices, values, and technologies in a particular local environment" [5], and the elements mentioned above may be constituted and perceived as an information ecosystem. The relationships occurring within the information ecosystem, and related to the flow of information as well as the effects that this flow causes, are crucial from the perspective of information balance, i.e., the way of perceiving and interpreting the surrounding reality. Information—regardless of the form: a visual, textual, or sound, is a set of meanings occurring at different frequencies, where a single meaning could be perceived as an aspect [6] or a single component of a unit of cultural transmission—meme [7].

Digital Visual Content (DVC) consists of various types of meaning components (attributes), which can be interpreted by the recipients in different ways, depending on simple users' demographics (age, gender, location) as described by Wu et al. [8] or generation name (Millennial, Gen X, Boomer, Silent) [9]. According to Bianchi-Berthouse and Lizetti [10], the stimulus (visual content) triggers the generation and further expression of recipients' emotions. As regards to travel-related content [11], such a problem is directly related to the process of distinguishing the attributes (aspects) of photographs and expressing an emotional response. Emotions arise as a result of mental activity related to the cognitive process and further processing of information coming from the environment [12]. Due to the possibility of expressing emotions of different nature (positive or negative) and the intensity of shared content [13]—such diversity can be described as affective polarity. At the same time, emotions constitute one of the factors that imply the activity of the recipients in relation to the shared content. In Facebook's case, expressing emotions in response to shared content is accomplished by using the affective options of the "like it" function [14], which affects also their range [15]. Emotions are shaping the way of expressing and positioning oneself in relation to the shared content [16].

The article discusses issues related to the visualization of the specificity of the archetypal image of a person performing a specific tourist activity, such as hitchhiking, taking into account gender differences. The archetype was created based on the semantic decomposition of visual content into single-aspect terms. [6].

Simultaneously, the frequency of expressing the emotions of content recipients was visualized in relation to the pool of aspects connected to the image of hitchhikers. This will allow for the creation of a basis for multidimensional analysis of the phenomenon of affective label polarity according to simple demographic characteristics (here: gender), and building a scientific framework for memetic content management.

## 2. Literature Overview

### 2.1. Hitchhiking

Hitchhiking is a form of tourism in which travel occurs thanks to the kindness of car and/or truck drivers, traveling for private and/or business purposes. According to Mahood [17], hitchhiking or thumb traveling is a specific form of self-expression reflecting personal identity. It is also seen as a form of slow, alternative, and sustainable mobility that aims to experience existential authenticity in tourism [18].

### 2.2. Aspects

Aspects are the characteristics of a specific subject or entity. They may include the semantic expressions of objects or phenomena occurring within the contents of a photograph and accompanying the main subject of a photograph or object. In the case of textual content, e.g., hotel reviews, the set of aspects may include: value, room, location, cleanliness, service business [19], i.e., direct semantic reference to hotel service components.

Kim and Stepchenkova [20] characterized the content of a photograph in the context of the presence of foreground meanings (objects). Simultaneously, Sion [21] analyzes body types (defined as separate meanings) most and least frequently represented in social media selfies, and positions the human body as an element of digital subjectivity.

These meanings, according to the typology proposed by Lu et al. [22], are mostly nouns or verbs (referred to as head) together with the accompanying, so-called modifiers (expressed in the form of adjectives or adverbs) which function as qualitative descriptors of nouns or verbs. Hence, cognitive attributes (qualitative labels, i.e., modern/traditional) can be attributed to the meanings within the content. In turn, in relation to the content, affective reactions are expressed.

Stylos et al. [23], from the perspective of tourist destination image and based on Baloglu and McCleary [24], Bigne et al. [25] and Hallmann et al. [26] stated that the affective component of an overall image refers to the emotional responses or appraisal of the individual, reflecting a tourist's feelings towards a destination. In this case, the destination

is expressed as a set of aspects based on perception and cognitivity, or tourist experience. Besides the basis, every single attribute is connected to a specific meaning. Machajdik and Hanbury [27] pointed at the role of high-level image semantics (descriptors) as the main elements containing and generating emotions. Simultaneously, Bianchi-Berthouze [28] created an image filtering system, which used low-level features (shape, textures, and those associated with verbalized human feelings and emotions. This is a crucial issue for the proposed research, where visualized meanings are combined with emotional and behavioral contexts. The assumptions presented above were supported by Cushman and McPhee [29]. The authors defined communication as a transfer of "symbolic information" in which "the source intentionally attempts to affect the receiver's behavior". Thus, the emotion expression is a kind of behavior towards acquired and processed content.

Based on the stated above, digital representations of people, objects, phenomena, etc., can trigger a varied emotional response. The polarity of emotions may be conditioned by the above-mentioned demographic characteristics (gender) and result from the existence of differences in the perception of products and services. [22]. According to the assumptions described by Yang et al. [6], ABSA (Aspect-Based Sentiment Analysis) methodology enables the identification of aspects, as well as the emotions expressed towards them.

### 2.3. Memes and Information Ecology

Memes are cultural replicators, i.e., cultural information carriers [7], and represent particular meanings. Vaneechoutte and Skoyles [30] defined memes as "bits of behaviorally transmissible information". In contrast, Sperber [31] described memes as "shared representations". Memes, like biological replicators [32], are spreading within the population of recipients through copying or imitation [33], penetrating from one mind into another one [34]. Memes can be described from three main perspectives [35]: 1. content (meanings which are transmitted), 2. form (form of content expressed: textual, graphical, other); 3. stance (relationship between the source of transmission, meme, and its recipients). Thus, memes as carriers of information, and according to communication theory [36], cause changes in the mental conditions of recipients and influence their activities, including the affective ones. Such type of changes could be perceived as a specific effect of the communication process on the arrangement: source-channel-message-recipient effect.

Nardi and O'Day [5] defined "information ecology" as "a local environment enriched with multiple heterogeneous technologies, such as personal computers, handheld devices, and interactive screens, which are interlinked as a unified system". Adding human factors, the main goal/role of such system, apart from providing solutions of problems, etc., is mainly a transfer of meaning (as memes that are sets of "aspects") expressed in textual, graphical, or other forms as well as to create and achieve specific responses to such content (e.g., behavioral). Keeping in mind the above, the more appropriate notion to describe such a framework will be the "information ecosystem", i.e., combination of human and computer factors, staying in specific relations directly connected with the content transfer and further effects of such transfer.

## 3. Scientific Problem Development

The assumption that memes as a set of aspects are present in visual digital content forming the archetype was done. Archetype is a primal idea of an object embedded in human experience [37]. The archetype triggers " . . . strong emotional responses in those who are exposed to them. The archetype concept is predicated on a person's previous emotional encounters and interactions with similar characters or ideas" [38].

The visualization of the frequency of occurrence of aspects in a meme is similar to the creation of their maps, which reflect the essence of a specific object or phenomenon. Concerning verbal meaning, i.e., transmitted in the form of text, DeLosh and McDaniel [39] suggested that there is a positive correlation between the use of specific expressions (meanings) in the source message and their frequent use and processing by the recipients. There is a probability that if the frequency of particular aspects in the content of the

photograph is high, then—in theory—the reactions of recipients related to the processing of specific content, including emotional reactions, may also be more frequent.

Assuming that memes are spread through imitation and copying, and the successfulness of a meme is based on its frequent aspects, together with the frequent generation of emotions, the following question becomes important:

RQ1: In the holistic perception of the archetype of a specific phenomenon as a meme, could the frequency of occurrence of aspect terms be correlated with the differentiated emotional response of content recipients?

Facebook, like other social platforms, enables users to interact in a virtual world on the basis of Secondary Social Bonds (SSB). These ties cover all manifestations of people's integration, also the ones beyond virtual reality, and have been defined as weak ties, lacking a strong emotional context, between group members clustered, among others, around common interests and related activities [40], and bear the marks of homophily [41]. These activities relate to communication, commenting, exchanging and sharing information—in text, graphics, and multimedia forms. Among the various content referred to as User-Generated Content, photos are particularly eye-catching and trigger emotions [42]. Simultaneously, Geurin-Eagleman and Burch [43] suggest, based on Marshall [44], that "social media has the ability to serve as a platform, on which an individual can build a public self-presentation", which can be synonymous to a transfer of memes.

Therefore, the collection of photographs shared by the users in a group based on secondary social bonds is not only a specific element of the nonverbal Visual Narrative Art (VNA)—the storytelling technique based on the presentation of the visualized representation of meaning [45]—it is also a holistic expression of an archetype (meme), constructed from meanings (aspects) by virtual communitarians [46]. The selection of meanings and their incidence are conditioned, in this case, by the influence of the surrounding social environment. In the case of brand management, Högström et al. [47] suggested that the subject independently chooses the attributes, based on that which he/she wants to build his/her image, taking into account the standards resulting from the environment, and the ability to build values on their basis. In the evolutionary context, it literally refers to the creation of an image, thanks to which it will be possible to achieve a competitive advantage and achieve specific goals. Highhouse et al. [48] suggest that by using appropriate content—i.e., by managing the elements of shared content, the source of a message can influence the perception of the subject (themselves, other people, groups, places, phenomena) and create appropriate emotional reactions of the recipients.

Hence, the next question was formulated (RQ2): Bearing in mind the secondary social bonds and the phenomenon of homophily, which type of emotions (positive or negative) and in what intensity will they occur in relation to the particular meanings of the hitchhiker archetype (meme).

## 4. Materials and Methods

The work was aimed at implementing the ABSA methodology within the framework of the meme theory and information ecology for the semantic decomposition and analysis of DVC. The subjects of the research were original photographs published between March and May 2018 by female hitchhikers (*n* = 150) and male hitchhikers (*n* = 150) within the closed group "Autostopowicze czyli My" [Us—Hitchhikers], functioning on Facebook. The group brings together people from Poland, who hitchhike and includes 47,557 people (as of 27 August 2018). Through a preliminary, subjective, and comprehensive analysis of the shared photos, the authors found that their content was similar from the perspective of many years. Therefore, only photos from one quarter of the year were selected for further consideration and analysis. Three hundred shared photos account for over 30% of their annual number, which allows for empirical research, The photographs as well as the behavioral activities of recipients were examined by using netnography—a method of researching online communities and providing information about their members, their behavioral patterns and activities [49], and quantitative analysis methods.

Within each photo, individual elements (aspects) were distinguished based on the assumptions of the decomposition of tourist photos proposed by Kim and Stepchenkova [20]. The original typology was adapted through an initial examination of hitchhiking photos, and the presence of four categories of meanings, referring to the characteristics of the main subject within the photographs was indicated:

- Hitchhikers' Surroundings—"other people" (other hitchhikers and/or locals), "my equipment", "transportation infrastructure", "city", "countryside", "landscapes" (both natural and cultural);
- Hitchhikers' Activities—"active rest", "passive rest", "car driving" (on a trip), "walking";
- Season—"spring/summer", "autumn/winter";
- Heritage—"architecture", "art", "local artefacts", "food", "country officials", "country attributes'.

The analysis of the image content was carried out in three main steps (Figure 1).

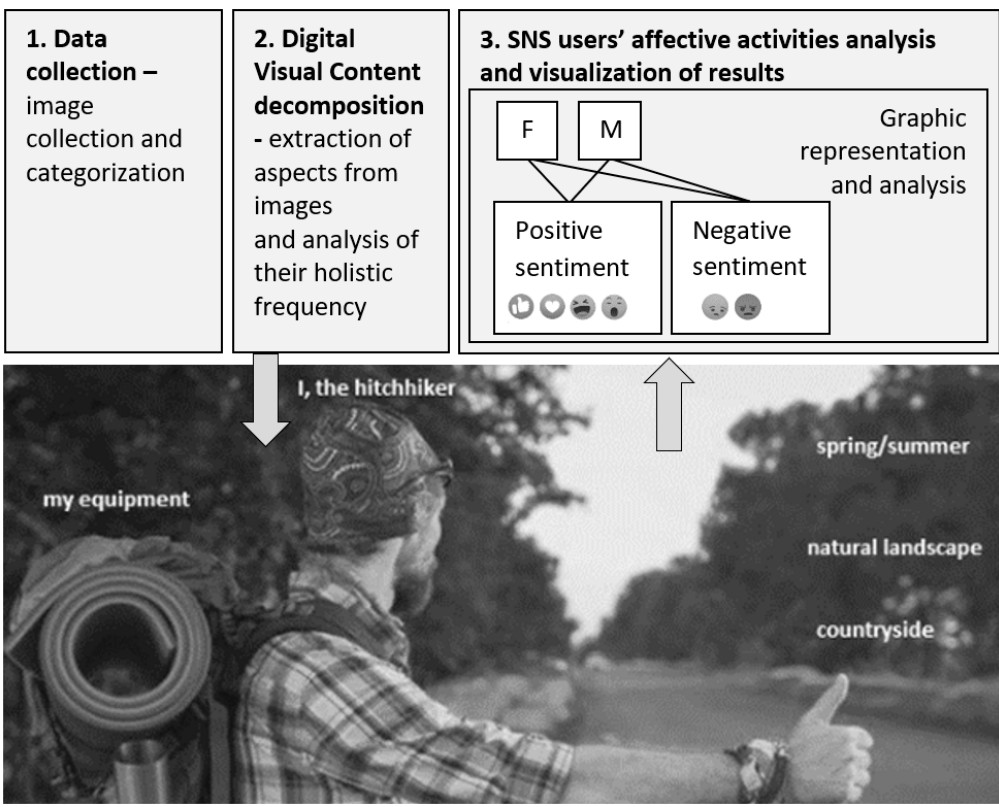

**Figure 1.** Theoretical construct of the study.

First of all, the content of the photographs shared by men and women was analyzed in the context of meanings defined by Kim and Stepchenkova [20]. The gender criterion results from differences in the way of SNS usage (in the context of self-presentation) and in the SNS activities of women and men [50]. In each of the photos shared by hitchhikers, features belonging to the four groups of aspects and those indicated above were searched. The results were recorded in Excel sheets.

This holistic analysis of separated groups of photographs led to the creation of two sets of aspect pools, where, for each category, its components were found in different frequencies. They also constituted a comprehensive visualization of the manner of self-presentation for a given gender of hitchhikers.

To find differences between the numbers of aspects issued within the analyzed photographs, the nonparametric Kruskall–Wallis test was used [51].

Secondly, data on the number of likes was collected (taking into account their diverse emotional characters [52], i.e., positive feedback (ike, love, wow, haha) and negative feedback (sad, angry). The obtained numerical data are arranged according to the gender criterion (F, M). In this case, the number of likes is not only a direct expression of interest in specific content but also allows to indicate the emotional context of this phenomenon [14].

In the third stage, following the work of DeLosh and McDaniel [39], it was assumed that the greater the frequency of a given aspect within the gender-defined pool, the more emotions will be expressed in relation to it. Similarly, in the model proposed by Wu et al. [8], one of its components was the analysis of the correlation between visual features (in the case of this study: aspect—verbalized visual attributes of the photo) and the accompanying emotions. These emotions (positive and negative) were expressed by choosing one of the six variants of "like it", i.e., like (acceptance), love (love), haha (joy), wow (excitement), sad (sad), and angry (angry).

The relationship between the total number of aspects isolated from the photos of male and female hitchhikers, and the diverse affective active interactions between women and men were also examined.

Spearman's Nonparametric, as well as Kendall's Tau (the analyzed variables related to the number of particular aspects had the characteristics of dichotomous variables, where "1"—the aspect is present; "0"—the aspect is absent) correlation tests [53] were used to analyze these relationships.

To unify and visualize the obtained results, all statistically significant correlations whose Tau value was greater than or equal to 0.2 (positive correlation) or −0.2 (negative correlation) were ranked as "1" and "−1" respectively. In the case when the value of the correlation coefficient was greater than or equal to 0.4 or −0.4, the rank was "2" and "−2" respectively. Statistically insignificant results were removed.

The obtained ranks were summed up depending on the type of emotion, i.e., positive or negative, expressed in relation to individual aspects. The obtained data were visualized in the form of radar charts. The charts were made with the use of the MS Excel software. Statistical calculations were carried out using the Statistica 13 suite.

## 5. Results

Among the 150 photographs shared by men, 146 photos of them were found to contain the subject of the photograph, i.e., the hitchhiker himself, surrounded by 723 various aspects ($\bar{x}$ = 4.9). In the case of female hitchhikers ($n$ = 150), the presence of the subject was demonstrated in 149 photographs. They were accompanied by 688 aspects ($\bar{x}$ = 4.6). The detailed distribution of the number of analyzed aspects is presented in Figure 2.

The results of the Mann–Whitney U test showed the existence of statistically significant differences between the mean values for the four variables within the hitchhikers' Surroundings (equipment, $p > 0.001$; landscape, $p > 0.01$; city, $p > 0.04$; countryside, $p > 0.01$); two within Season (spring/summer, $p > 0.04$; autumn/winter, $p > 0.04$) and one variable within Heritage (country attributes, $p > 0.02$). In the case of other variables, there were no similar differences between the mean values of variables for the groups distinguished based on analyzed demographic characteristics (gender). Individual aspects within the analyzed DVC met with a diverse emotional reaction of the recipients: men generated 8202 affective active interactions, while women generated 6704. Details, along with basic descriptive statistics are presented in Table 1.

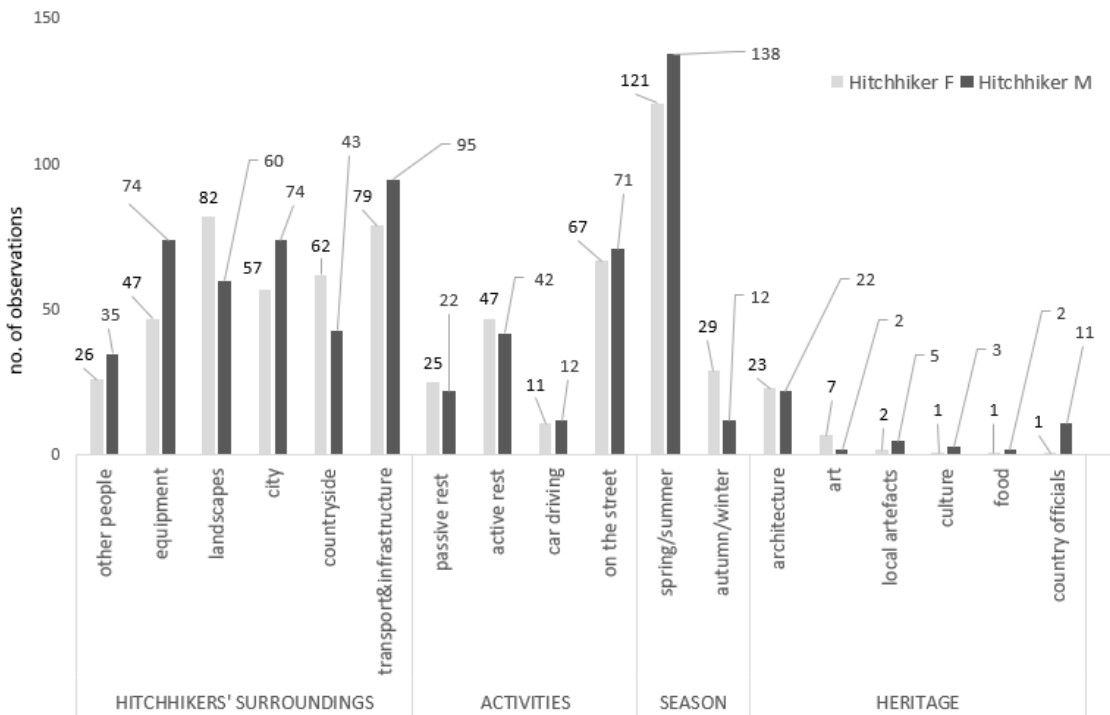

**Figure 2.** Distribution of the incidence of aspects in the content of images shared by hitchhikers (*n* = 300).

**Table 1.** Frequency distribution of different emotional reactions, based on the "like it" button, towards photographs shared by females (Hitchhiker F) and males (Hitchhiker M).

| | | Hitchhiker F | | | Hitchhiker M | | |
|---|---|---|---|---|---|---|---|
| | | No. of obs. | $\bar{x}$ | SD | No. of obs. | $\bar{x}$ | SD |
| Female affective active interactions | f_like | 2303 | 15.35 | 24.15 | 4259 | 28.39 | 99.34 |
| | f_love | 373 | 2.49 | 4.48 | 854 | 5.69 | 0.12 |
| | f_haha | 48 | 0.32 | 1.16 | 327 | 2.18 | 0.00 |
| | f_wow | 18 | 0.12 | 0.84 | 22 | 0.15 | 0.00 |
| | f_sad | 12 | 0.08 | 0.75 | 1 | 0.01 | 0.00 |
| Male affective active interactions | m_like | 2918 | 19.45 | 24.94 | 2304 | 15.36 | 46.08 |
| | m_love | 313 | 2.09 | 2.83 | 236 | 1.57 | 5.55 |
| | m_haha | 50 | 0.33 | 1.13 | 175 | 1.17 | 5.36 |
| | m_wow | 31 | 0.21 | 0.78 | 20 | 0.13 | 0.51 |
| | m_sad | 8 | 0.05 | 0.32 | 4 | 0.03 | 0.2 |

Source: Own elaboration of the use of the Statistica 13 suite.

Based on the results of nonparametric correlation analyses, the phenomenon of affective label polarity was visualized. In the case of reactions generated by women towards the photos of other female hitchhikers, a positive relationship between the analyzed aspects and the generation of positive emotions occurs for aspects of the hitchhiker's Surroundings group (landscapes, countryside, car driving and culture). In the context of the negative expression of emotions, a similar relationship can be noticed in the case of Heritage (country attributes).

In the case of photographs taken by men, the positive emotional response concerns the aspects of the hitchhiker's Surroundings group (other people, countryside) and the hitchhiker Activities group (car driving). In the case of both types of image sources, a number of negative correlations are also observed, i.e., the higher the incidence of a

particular meaning, the lower the probability of the occurrence of an emotional response. Details are shown in Figure 3.

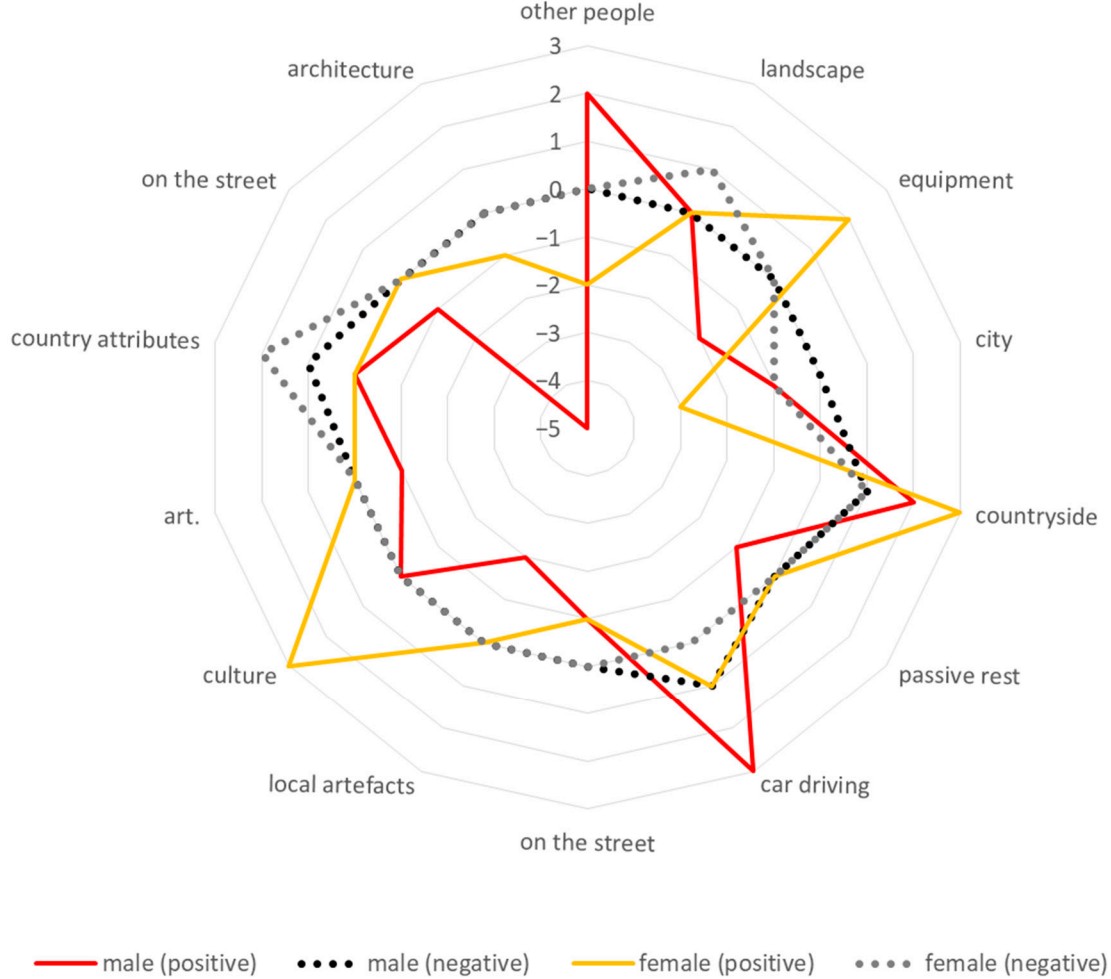

**Figure 3.** Female affective active interactions with images shared by females and males based on nonparametric correlation coefficient analysis.

In the case of men's affective activity (Figure 4), it was shown that in relation to photographs taken by women, there are statistically significant, positive correlations between the expression of positive emotions and their presence in the content of aspects related to: countryside, hitchhikers' activities (car driving, on the street) and a warm season (spring/summer). The decisively bigger variation of the frequency of expressing positive emotions concerns the photos taken by men. A positive correlation was observed between their expression and the aspects directly related to the category of hitchhikers' Surroundings (other people, countryside), Activities (car driving), Season (spring/summer), and Heritage (local artefacts, culture, food, and country attributes).

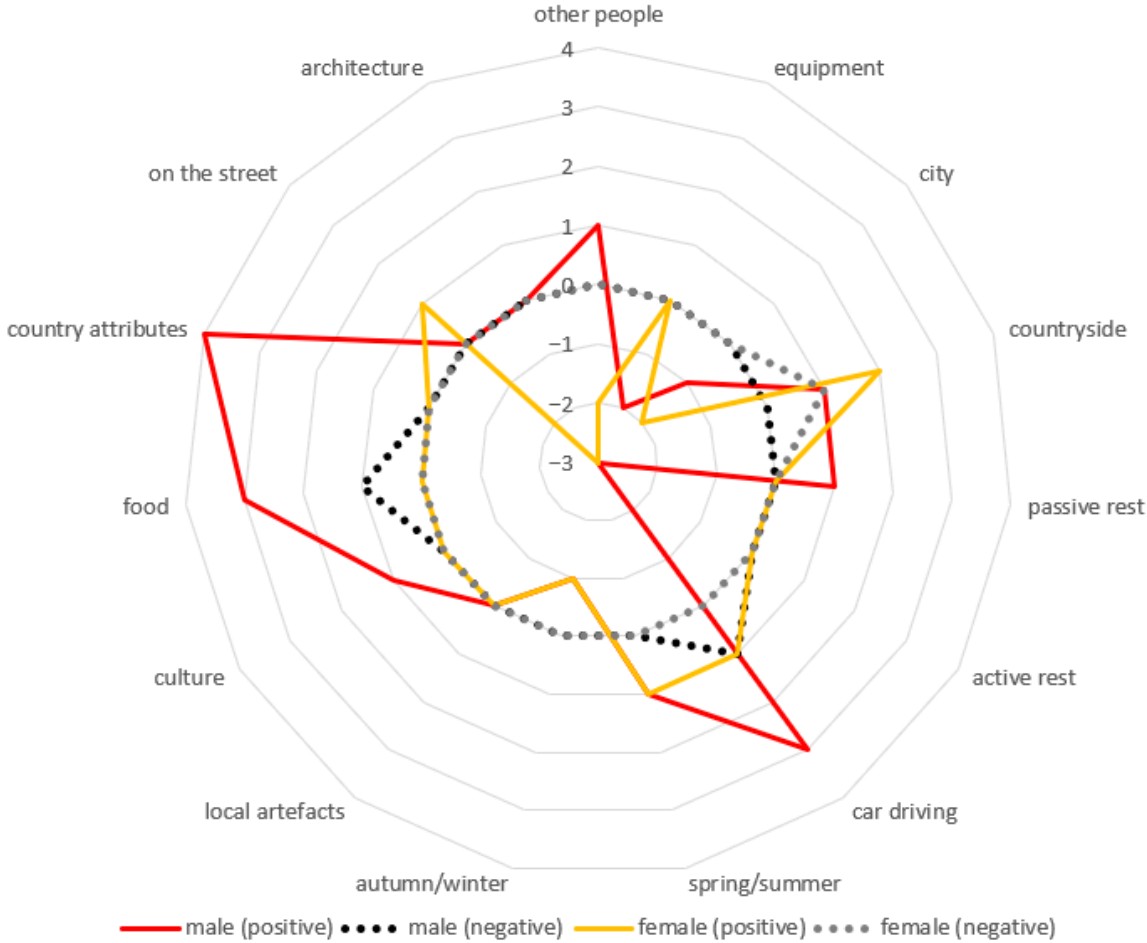

**Figure 4.** Male affective active interactions with images shared by females and males based on nonparametric correlation coefficient analysis.

## 6. Discussion

According to the obtained results, in specific thematic groups on Facebook, positive emotions related to shared content are probably expressed and shared more often than negative ones. Such a statement is also consistent with the results achieved by Vermeulen et al. [15]. The authors pointed to the important role of SNS' technical affordances in the transfer of emotions, as well as the significance of the emotional factor for the general cognitive process. It should be noted that in the proposed approach, positive emotions are probably expressed not towards the hitchhiker archetype, but directly towards its individual descriptors (individual aspects).

The dominance of positive emotions within homophilic groups is implied, among others, by the extensive support from peers who have similar experiences. Peers usually emanate positive emotions, are helpful, provide support, and share their own experiences [54]. On the other hand, such type of high experience could be described as a familiarity—a degree of acquaintance with situation experience [55]. Thus, the expression and location of the emotional factor, related to shared experiential digital content, may be slightly different within Facebook groups not characterized by homophily.

At the same time, it should be noted that, in the presented research, the transfer, processing, and expression of the hitchhiker archetype takes place on the mesoscale, i.e., in a relatively small group of people. There is a high probability that, in the case of groups that are heterogeneous in terms of interest and activities, or within the scope of the entire population (macro-scale), emotions would be expressed towards other visual features, as well as their nature and frequency being different.

The results of the study also fit into the 6A content management model [56], especially in the context of the second component, i.e., abridgements related to data capturing and its further extraction, as well as data mining and data visualization. Analyses of this type are also related to the achievement of various objectives, including drawing the audience's attention to specific content and increasing the level of their involvement with the message, as well as stimulating specific affective activities (component 1 of the 6A model—Activity sources i.e., management of SNS' user activities). This allows for the simultaneous construction and direction of a message to a targeted group of recipients. The effect will be the formation of a specific way of perceiving reality by the recipients. It may also refer to the personalization of the offer of products and services. This type of research is a part of the concept of social content management (SCM). SCM was defined by Glazkov [57] as a set of concepts, methodologies, and standards, which enable and facilitate the creation, organization, and maintenance of content through social interaction of individuals online. The decomposition of digital content and the categorization of the obtained descriptors can lead to the creation of effective content strategies aimed at shaping engagement of the content audience. It will also be useful for the typology of DVC aspects as elements of various content strategies, and to analyze the descriptors' effectiveness in achieving the goals of the SNS communication process, as mentioned by Juntunen et al. [58].

The role of the defined archetypes (memes) is, in this case, to create a conceptual and usable framework for a specific phenomenon, which enables defining the areas of their potential variability. Such variability is probably conditioned, among others, based on demographic attributes, by the diverse emotional perception of a meme and its components. A similar approach, in the context of characteristics of the human body acting as elements of digital subjectivity, was presented by Sion [21].

The spreading of carriers of cultural information within a population of recipients is similar to the gene transfer in a population located at specific ecosystems. The most popular memes (acting as sets of aspects) functioning as a holistic being—like phenotype conditioned by a specific gene arrangement in dynamically changing environments—are the subject of cultural selection—a mechanism similar to natural selection. Uploading and sharing photos as well as tracking audience reactions is an integral activity of users and, regardless of the type of activity, can be seen as an integral part of the modern lifestyle. In the similar context of specific cultural adaptation to modern technologies [59], the phenomenon of sharenting (sharing photos of children by parents) was described by Atwell et al. [60].

## 7. Conclusions

The article proposes a concept of netnographic methodology for analyzing the label polarity phenomenon connected with positive and negative emotions expressed in relation to decomposed DVC.

Within the group functioning based on secondary social bonds, i.e., the homophilic group, the gender demographic attribute may probably have a significant impact on the perception of individual aspects as well as the expression of emotions related to the aspects within the DVC. In turn, aforementioned SSB were developed based on both practicing tourism and belonging to a community, simultaneously affecting the manner and form of self-presentation as well as perception and processing information about other group members.

The study has a visible utilitarian character. Its results allow for the effective creation and management of DVC in the context of the affective behavioral actions of content recipients. The innovation of the work results from the application of the information ecology framework, as well as the memetic approach. After decomposing a randomly selected pool of photographs, a holistic set of aspects was created concerning the selected research object, creating a meaningful archetype of a hitchhiker. The least frequent aspects—directly related to the essence of the tourist experience—within the pool expressing the

importance of the archetype, most likely has a significant impact on the perception and appearance of an emotional response of the recipient. Such results lead to several practical implications for DVC development and its impact on the digital subjectivity of recipients to various socio-demographic profiles. It is also important to create and manage visual content to evoke the desired emotional reactions, specific behaviors, and attitudes of recipients resulting from the consumption of the content.

At the same time, it was shown that the proposed methodology may be used to effectively analyze the emotional response of the recipients to the shared visual digital content.

The limitation of the presented results consists mainly in the adopted qualitative analysis where the greater number of aspects, the greater emotional response is observed. To verify the results obtained through netnography, research using traditional questionnaires and interviews should be conducted, especially focusing on cognitive and emotional factors connected with the manner of DVC processing by recipients. The results of such queries should also be useful for the recognition of how users (within a homophilic group with SSB) perceive the photos shared in SNS, i.e., what elements of the image(s) draw recipients' attention and is there any relationship between the recipients' perception and emotional reaction(s)?

Further research should also include a multidimensional analysis of the undertaken problem, i.e., taking into account the remaining demographic attributes (i.e., age, education, place of residence). It would also be important to carry out experiments on large samples (>3000 photos), confirming the hypothesis about the impact of small levels of incidence on graphical representations of aspects of the expression of positive emotions.

**Author Contributions:** Conceptualization, K.S.; methodology, K.S.; formal analysis, K.S. and A.S.; investigation, A.S.; resources, K.S.; data curation, A.S. and K.S.; writing—original draft preparation, K.S.; writing—review and editing, K.S. and A.S. All authors have read and agreed to the published version of the manuscript.

**Funding:** This research is supported by the Bialystok University of Technology and financed by a subsidy provided by the Minister of Science and Higher Education, grant number WZ/WIZ-INZ/2/2019.

**Institutional Review Board Statement:** Not applicable.

**Informed Consent Statement:** Not applicable.

**Data Availability Statement:** The data presented in this study are openly available Mendeley Data at doi, reference number 10.17632/sp2wpmytsx.1.

**Conflicts of Interest:** The authors declare no conflict of interest.

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
