# Peer review of "Hitchhiking Experiences and Perception of Affective Label Polarity in Social Networking Sites—Potential Memetic Implications for Digital Visual Content Management"

_sustainability, doi:10.3390/su13010223_

Round 1
Reviewer 1 Report
Thank you for this improved, revised version of your paper, and for your clear revision statement. I now have a better understanding of the main methodological point of the study, and the benefits (and limitations) of the approach. This contribution can stand on its own, even if certain assumptions about meaning and memes I disagree with.
Author Response
Dear reviewer, thank you for your comments and contribution to improving the scientific quality of our article. The paper has been checked again. Editing, grammar, linguistic and punctuation errors have been corrected (changes marked in gray).

Reviewer 2 Report
The quality of the pictures must be improved.
Author Response
Dear Reviewer, thank you for your comments and contribution to improving the scientific quality of our article. We've made corrections to all drawings and photos to make them more visible and legible to readers.

This manuscript is a resubmission of an earlier submission. The following is a list of the peer review reports and author responses from that submission.
Round 1
Reviewer 1 Report
This paper was very difficult to follow, and the issue was not clarity of English. There was an astonishing amount of meta-level discussion of terminology throughout, much of it repeated and meshed together in a way that made comprehension—even for a scholar who works on memes and digital culture—extremely difficult. The actual topic of study—hitchhiker photos on social media—and the relevant background in tourism and leisure studies, is almost entirely neglected. The concept of meme deployed—as gene-like imitation, at the level of decomposed “aspect”—is highly idiosyncratic and certain discordant with the meme-culture literature cited. There were many, many errors of all kinds at the level of copy, but again my evaluation is not based on that fact. The issue, instead, is legibility—the sentence-to-sentence progression of meaning just isn’t there. Instead, there is a hodgepodge of methodological statements, sometimes delivered in acronyms, and very little that an informed reader can make sense of.
Reviewer 2 Report
Please, check the attach file.

Reviewer 3 Report
It is an interesting investigation due to its originality relative to the applied sample.
They should explain and state the sampling error. A quantitative study of a group of FBs with more than 47,000 members is carried out, so including the sampling error is basic.
Another interesting aspect would have been the inclusion of qualitative aspects, such as analysis of the discourse of the published texts.
Reviewer 4 Report
Important theoretical and methodological specifications must be made in order for the paper to be clearer and its argument more convincing. The manuscript could be markedly improved by clearly telling the reader what the theoretical, empirical and practical implications of the research might be. In the introduction section, the motivation and objective of the paper shall be further elaborated/discussed. There is a need of separating discussions that outline the research methodology with the empirical findings for the sake of clarity. The structure is weak and makes following the author's line of thinking a challenging task. A more discursive, analytical conclusion is needed, that engages with the theoretical questions in scholarship raised earlier in the paper. The conclusion should clarify the main contribution of the paper and the value added to the field. Some bibliographic references are simply brought up without being developed, or without an adequate explanation as to why they are relevant. The discussions require more structure and there is a need of offering a clear assessment of reviewed literature. Several statements made in the paper are not supported by adequate empirical evidence or by making reference to relevant literature.
The proportion of recent peer-reviewed published sources is quite low, and thus more recent references from Scopus- or WoS-indexed journals are needed. Here are some research suggestions that complement your approach (I am not the editor of these journals, member of the board, or author/reviewer):
Sion, Grațiela (2019). “Self-Portraits in Social Media: Means of Communicating Emotion through Visual Content-Sharing Applications,” Linguistic and Philosophical Investigations 18: 133–139. doi:10.22381/LPI1820199
Mircica, N. (2020). “Restoring Public Trust in Digital Platform Operations: Machine Learning Algorithmic Structuring of Social Media Content,” Review of Contemporary Philosophy 19: 85–91. doi:10.22381/RCP1920209
Sion, Grațiela (2019). “Constructing Human Body as Digital Subjectivity: The Production and Consumption of Selfies on Photo-Sharing Social Media Platforms,” Review of Contemporary Philosophy 18: 150–156. doi:10.22381/RCP1820199
Sion, Grațiela (2019). “Is Selfie-Posting Behavior a Kind of Nonpathological Narcissism?,” Analysis and Metaphysics 18: 71–77. doi:10.22381/AM18201910
Atwell, Gary J., Eva Kicova, Ladislav Vagner, and Renata Miklencicova (2019). “Parental Engagement with Social Media Platforms: Digital Mothering, Children’s Online Privacy, and the Sense of Disempowerment in the Technology-Integrated Society,” Journal of Research in Gender Studies 9(2): 44–49. doi:10.22381/JRGS9220193